# Change in address in electronic health records as an early marker of homelessness

**Janet Song** [1,2*], **Yu Deng**[1,2,3], **Yuyang Yang**[1,2], **Lacey Gleason**[1,2], **Abel Kho**[1,2,4]

**1** Feinberg School of Medicine, Northwestern University, Chicago, Illinois, United States of America,
**2** Center for Health Information Partnerships, Northwestern University Feinberg School of Medicine,
Chicago, Illinois, United States of America, **3** AbbVie Inc., Chicago, Illinois, United States of America,
**4** Division of General Internal Medicine, Department of Medicine, Feinberg School of Medicine,
Northwestern University, Chicago, Illinois, United States of America

\* Janet.song80@gmail.com

## Abstract

### Introduction

Housing stability is a key health determinant and there is a need for early screening for instability with existing electronic health record (EHR) data to improve health outcomes. We aim to establish recorded address changes as a screening variable for housing instability and homelessness and to attempt to define the threshold of high churn.

### Methods

Our study is a single-center cross-sectional study of EHR data (2018-2024) conducted at a US academic center with eleven sites across Chicago. We include patients 18 years or older with at least three hospital encounters over three different years. We define address churn as the number of address changes recorded in the EHR corrected to three-year intervals. We compare demographic and clinical characteristics of individuals with varying address churn with the student T-test to look at distribution of address churn for patients with and without record of homelessness, ANOVA to evaluate the distribution of ages for different levels of churn, and the chi-square test to evaluate for association between churn and clinical diagnoses. We perform multivariable logistic regression to measure the association between people with a record of homelessness and address changes.

### Results

The study includes 1,068,311 patients with 756,222 having zero address changes, 156,911 having one address change, 137,491 with two address changes, 9,558 with three address changes, and 8,129 with four or more address changes. People with no record of homelessness in the EHR have mean address changes of 0.6 (SD 0.7) whereas people with record of homelessness have mean address changes of 1.8 (SD 1.3). Diagnostic profiles of the varying address change groups show increased prevalence of psychiatric diagnoses (65.2% in the 4 or more-address change group) compared to lower address change (27.7% in the 0-address change group). Address churn is significantly associated with homelessness with an odds ratio (OR) of 1.44 (95% CI = [1.42-1.47], P < 0.001).

**Data availability statement:** Data cannot be shared publicly due to legal restrictions as data contain potentially identifying patient information. Data are available from the Northwestern University Northwestern Medicine Enterprise Data Warehouse (NMEDW) (contact via fsm-research@northwestern.edu) for researchers who meet the criteria for access to confidential data.

**Funding:** The author(s) received no specific funding for this work.

**Competing interests:** I have read the journal's policy and the authors of this manuscript have the following competing interests: Dr. Abel Kho is an advisor for Datavant. This does not alter our adherence to PLOS ONE policies on sharing data and materials.

## Conclusion

Our results support a role for residential address churn in screening for housing instability in healthcare systems and reinforce the association between psychiatric disorders and housing instability. Our findings can help public health policy makers in targeting vulnerable populations at risk of homelessness with multiple health comorbidities for housing interventions.

## Introduction

Homelessness was estimated to affect 582,462 people in the United States in 2022, disproportionately impacting individuals identifying as Black or African American [1]. Historically, homelessness has been influenced by economic downturns, housing policies, and social safety nets, with significant spikes during periods such as the Great Depression and the post-industrial era [2]. Recent trends indicate a rise in chronic homelessness, exacerbated by the COVID-19 pandemic [3].

Some consider homelessness as one extreme of the housing stability spectrum with stable housing on the other end [4]. Housing instability encompasses a range of challenges including trouble paying rent, overcrowding, frequent residential moves, or spending a large portion of one's income on housing. These issues are intertwined with broader social determinants of health, such as education, employment, and social support networks [5].

The landscape of homelessness research has evolved, highlighting the critical need for evidence-based policy responses. Initiatives such as Housing First and supportive housing programs have emerged as effective strategies to address chronic homelessness by offering stable housing without preconditions [6–8]. However, the implementation of these programs often faces significant barriers, including funding limitations, policy inconsistencies, and systemic challenges that hinder effective outreach and support for the homeless population [9–11].

Homelessness and housing instability greatly impact health and healthcare costs. They have been linked with increased psychiatric disorders, substance abuse, cardiovascular disorders, infectious disease, and all-cause mortality [12–15]. Homelessness is also associated with increased emergency department (ED) utilization and longer hospital stays [16,17]. While research is ongoing, the link between housing and health appears multifaceted and intersects with issues such as accessibility and social support [18]. Effective screening for housing instability can help reduce incidence of unmanaged disease, decrease healthcare costs, and uphold patient-centered care [19].

### Related works

Current methods using electronic health record (EHR) based screening tend to miss patients experiencing homelessness [20]. The International Statistical Classification of Diseases and Related Health Problems: 10th Revision (ICD-10), Z-codes were first made available in the 2016 fiscal year to track social determinants of health in patient records [21]. However, usage of these codes has been infrequent, as low as 2.03% of patients with Z-codes recorded [22]. Other strategies include adding structured fields to capture social determinants of health (SDOH) but similarly showed low utilization with less than 1% of encounters having structured fields completed [23]. While providers often recognize the importance of addressing SDOH, they report multiple barriers in recording or retrieving the psychosocial history such as discomfort discussing sensitive topics with patients, limited time in clinic, and logistical difficulties retrieving data from EHR [24,25].

Screening questionnaires tend to be more time intensive and miss a significant portion of homeless individuals [26]. Integrating screening with intervention has shown great potential and praise from patients but often is difficult for providers and resource intensive for health systems [27,28]. As a result, follow-up for patients who are identified is often poor [29,30].

Natural language processing (NLP) in EHRs has become an increasingly popular method to scan notes to identify patients experiencing homelessness but has yet to make its way into real-world practice [31–33]. NLP model accuracy is only as good as how well SDOH are documented in the EHR and is often not generalizable as the training occurs with a specific health system and patient population [34–36]. Furthermore, these methods identify patients who are already homeless, not those at risk or on the verge of homelessness. Developing a social risk score is promising as it would use data that does not rely on providers such as Z-codes and would not suffer in accuracy depending on free text quality as in NLP, but current performance is not yet acceptable as there is a need to explore more variables to create a more robust score [37]. Our study attempts to fill in the gaps in SDOH screening in health systems by exploring frequency of address changes, an unbiased variable that is readily available and regularly recorded at the time of intake.

While a variety of screening tools and methods exist using EHR data, housing stability remains difficult to track. We hypothesize that frequency of address changes, or address churn, is a marker of housing instability and is associated with poor health outcomes. We evaluate our hypothesis by running a multivariate logistic regression analysis with data available in the EHR to test if there is an association between address changes and homelessness. Given that address changes are poorly described in the literature, our study also seeks to explore how diagnoses vary with different levels of churn. We answer this question by performing a chi-square analysis for diagnoses and address churn. We also explore how demographic characteristics vary with address churn. We answer this question using mixed methods, combining statistical subgroup analyses and qualitative geospatial analysis. This comprehensive analysis of address churn can help fill in the gaps for patients who are not asked about their living situation or whose narrative is obscured in the EHR.

## Methods

### Data and study population

We utilize a retrospective cross-sectional study design to get a baseline snapshot of address churn using Northwestern Medicine's (NM) Electronic Data Warehouse (EDW) EHR data including the dates 1/1/2018-1/1/2024. The NM-EDW is a data repository that includes clinical and research data from sites within Northwestern University School of Medicine and Northwestern Medicine Healthcare (NMHC) for greater than 10 million patients spanning from 1970 to the present. It contains demographic data (age, gender, race, address), clinical data (diagnoses, medications, encounters, procedures), and outcomes extracted from the Northwestern Medicine EHR. The data was accessed for research on 3/5/2024. Authors had access to MRNs, which could identify participants, after data collection. We include adult patients who were at least 18 years of age as of 1/1/2018. To better capture residential address mobility over time, we include patients with three or more encounters spread across three different years. This results in 1,068,311 patients included in our study. We use structured data from the EHR including diagnosis recorded as ICD-10 codes, residential address, insurance provider, age, gender, and race. Gender and race are self-reported and extracted as recorded in the EHR. Data on race was collected due to the complex nature of race and housing [38]. The race group "Other" in this study includes patients who declined to report race or recorded race of "unknown". The study protocol was approved by the Northwestern University

Institutional Review Board. Waiver of informed consent was obtained for this study because the research involved no more than minimal risk to patients as this study uses pre-existing data, the waiver will not adversely affect the rights and welfare of participants, and it would not be practicable to carry out the research without the waiver.

## Address standardization

Residential address is input in a free text format at the time of patient intake and may be revised at check-in for follow-up appointments and admissions. Date and time of the residential address record are included as metadata. The free text nature of the residential address results in spelling variations which creates an additional address record. We process residential addresses using the United States Postal Service (USPS) Postal Addressing Standards. The USPS Postal Addressing Standards is a comprehensive set of guidelines that ensure proper address formatting for efficient mail processing and delivery [39]. We first convert all address text to uppercase. We create dictionaries of different abbreviations and common spelling errors for street suffix (i.e., Ave, Avenue, Av, Avn), directional indicators (north, south, east, west), and state names mapped to the standard USPS abbreviation. Standardized formatting for address spelling and capitalization allows us to decrease overcounting of address changes for more accurate calculation of address churn as described below.

## Identifying individuals who experienced homelessness

We consulted with housing workgroup experts to identify patients who have experienced homelessness. With the housing workgroup's guidance, we developed the following initial criteria to find patients experiencing homelessness which included ICD-10 code of Z59.0, residential address that either indicates "undomiciled" or "homeless", congregate living facility as home address, or hospital address as home address [32,40].

We conducted a blinded chart review of 50 randomly selected patients, stratified by housing status with 10 that met criteria for homelessness and 40 patients without criteria for homelessness. Among the 40 patients who did not meet the criteria for homelessness, we found no evidence of any previous experiences of homelessness in their medical charts. Of the 10 patients who met the criteria for homelessness, only 5 had documented evidence of homelessness in their charts. The remaining 5 patients, despite meeting the criteria for homelessness, did not have documentation of homelessness in their charts. Instead, these 5 patients were identified as homeless based on their use of the hospital address. Based on the results from our chart review, we were initially overcounting homelessness by including patients who have the hospital as their address. We remove this criterion and conduct our study based on the final criteria: ICD-10 code of Z59.0, residential address that either indicates "undomiciled" or "homeless", and congregate living facility as home address.

Table 1 provides a summary of patients included in our study with approximately 0.2% of our study population having record of homelessness.

## Characterizing and defining address churn

We define address churn as the number of times a person's residential address changes over time. In this study, we quantify address changes corrected to three-year intervals to standardize the data across different time spans of patient encounters. For instance, a patient with encounter data spanning six years with six instances of address changes is adjusted to an average of three address changes per three-year interval. Address churn is calculated by counting each instance the patient's address record changed. Addresses are standardized as above to decrease overcounting address record changes. Here, we capture the demographic

**Table 1. Baseline demographic variables.**

| Group | 0 Address Churn (N = 756,222) | 1 Address Churn (N = 156,911) | 2 Address Churn (N = 137,491) | 3 Address Churn (N = 9,558) | 4 or More Address Churn (N = 8,129) |
|---|---|---|---|---|---|
| Age Mean (SD) | 52.1 (16.5) | 46.2 (17.2) | 41.9 (16.9) | 38.1 (15.9) | 37.9 (16.0) |
| Sex | | | | | |
| Female Count (%of group) | 472,782 (62.5%) | 101,446 (64.7%) | 92,164 (67.0%) | 6,784 (71.0%) | 5,957 (73.3%) |
| Male Count (% of group) | 283,440 (37.5%) | 55,465 (35.3%) | 45,327 (33.0%) | 2,774 (29.0%) | 2,172 (26.7%) |
| Race (% of group) | | | | | |
| White | 505,766 (66.9%) | 107,414 (68.5%) | 95,711 (69.6%) | 6,663 (69.7%) | 5,656 (69.6%) |
| Black or African American | 39,420 (5.2%) | 13,341 (8.5%) | 14,617 (10.6%) | 1,293 (13.5%) | 1,223 (15.0%) |
| Asian | 24,179 (3.2%) | 6,809 (4.3%) | 6,375 (4.6%) | 398 (4.2%) | 315 (3.9%) |
| AI/AN/NHPI | 1,622 (0.2%) | 463 (0.3%) | 450 (0.3%) | 30 (0.3%) | 38 (0.5%) |
| Other | 185,235 (24.5%) | 28,884 (18.4%) | 20,338 (14.8%) | 1,174 (12.3%) | 897 (11.0%) |
| Insurance (% of group) | | | | | |
| Commercial | 402,656 (53.2%) | 93,449 (59.6%) | 88,421 (64.3%) | 6,464 (67.6%) | 5,187 (63.8%) |
| Medicaid | 35,402 (4.7%) | 11,491 (7.3%) | 12,227 (8.9%) | 1,048 (11.0%) | 1,119 (13.8%) |
| Medicare | 233,307 (30.9%) | 37,248 (23.7%) | 25,742 (18.7%) | 1,440 (15.1%) | 1,375 (16.9%) |
| Other | 84,857 (11.2%) | 14,723 (9.4%) | 11,101 (8.1%) | 606 (6.3%) | 448 (5.5%) |
| ADI Mean (SD) | 0.30 (0.09) | 0.31 (0.09) | 0.31 (0.08) | 0.32 (0.08) | 0.32 (0.08) |
| Record of Homelessness (% of group) | | | | | |
| No record | 755,780 (99.9%) | 156,567 (99.8%) | 136,660 (99.4%) | 9,440 (98.8%) | 7,847 (96.5%) |
| Positive Record | 442 (0.1%) | 344 (0.2%) | 831 (0.6%) | 118 (1.2%) | 282 (3.5%) |

AI/AN/NHPI = American Indian/Alaskan Native/Native Hawaiian and Pacific Islander.

variances for different levels of address churn. We plot the distribution of address churn as a bar plot for people who have record of homelessness in the EHR and people who have no record of homelessness to establish a threshold for high address churn. To understand address churn in the context of Chicago neighborhoods, we map the geospatial distribution of churn. We calculate the mean address churn per zip code, plot it on a map of Chicago from the U.S. Census Bureau, and compare it to a map of area deprivation index (ADI) per zip code [41]. ADI accounts for several measures of SDOH and is normalized to a scale of 0 to 1 with greater values corresponding to greater deprivation and lower socioeconomic standing [42].

Address churn is categorized into groups of 0 address changes, 1 address change, 2 address changes, 3 address changes, and 4 or more address changes to compare prevalence of diagnoses and age and gender distributions.

## Processing diagnosis data

For the multivariate regression analysis, we code whether a patient has ever had a diagnosis of severe mental illness or substance abuse as these have been shown to be risk factors for experiencing homelessness. We use the Healthcare Cost and Utilization Project's (HCUP) Level 2 groupings for behavioral health diagnoses to create more meaningful groups and include ICD-10 codes from groups beyond F00-F99 pertaining to behavioral health [15,43].

To address our question of how diagnoses change with varying levels of address churn, we tabulate diagnosis frequencies by churn group and identify the top ten diagnoses. We group diagnoses from the top ten into cardiovascular diagnoses (essential hypertension, hyperlipidemia, and pure hypercholesterolemia), psychiatric diagnoses (anxiety and fear-related disorders, depressive disorder, and anhedonia), other signs and symptoms (other signs and symptoms, other abnormal glucose, vitamin D deficiency, and weakness), infectious

symptoms and diagnoses (cough, acute respiratory infection unspecified, and acute pharyngitis unspecified), and pain-related diagnoses (knee pain and other chronic pain). An age-stratified chi-squared test assesses the association between diagnoses and churn levels. Age groups include young adult (18-35 years), adult (35-65 years), and senior (65 + years) to account for any between age group differences in diagnoses.

### Confounding variables

As we evaluate the association between homelessness and address churn, we account for several confounding variables. Insurance status, which can act as a confounder and a proxy for economic status, is adjusted for by including it in our model [44,45]. Neighborhood factors, which likely influence housing status, are approximated using the ADI and are corrected for by incorporating it into our model. Mental illness, another potential confounder, is controlled for by including it in the model. Further details of the model are provided in the next section.

### Statistical analysis

To test our hypothesis that there is an association between the history of homelessness (binary outcome) and address churn, we use multivariate logistic regression. Address churn is the predictor variable of interest and is an integer ranging from 0 to 11. We run both univariate and multivariate models, adjusting for relevant demographic and clinical variables including age, self-reported gender and race as recorded in the EHR, insurance status, mean ADI of a patient's collective addresses, any diagnosis of severe mental illness (bipolar, schizophrenia, or psychosis), and any diagnosis of substance use disorder. We fit a logistic regression model using the statsmodels module in Python. All analyses are completed in Python v3.6.

To answer the question of how patient demographic characteristics varied with different levels of address churn, we utilize various statistical tests. We run a student T-test to compare distributions of address churn for patients with record of homelessness and those without. We evaluate how age and address churn vary with an ANOVA comparing age distributions across different levels of address churn. The statsmodels module is used to run the analyses listed above.

## Results

### Demographic characteristics of address churn

We observed significant differences in demographics across levels of address churn. The majority of the study population did not move over three years. Patients with a record of homelessness showed a mean churn of 1.83 (SD 1.34), significantly higher than patients without homelessness records, who had a mean churn of 0.58 (SD 0.71) (Fig 1A). A student T-test demonstrates that the difference is significant (p-value < 0.001). The distributions of age and gender differ across the levels of churn. The mean age decreases across the varying churn groups from a mean age of 52 in the zero-churn group to a mean age of 38 in the four or more-churn group (Fig 1B). A one-way ANOVA compares the age distribution across four groups, revealing a statistically significant difference in age distribution between the groups. Post hoc Tukey test reveals that age across groups was significantly different (p < 0.001 for all pairwise comparisons).

### Diagnostic characteristics of address churn

We find diagnoses differ across the different levels of address churn. A chi-squared analysis looking at the five diagnostic groups and the five churn groups finds diagnosis is

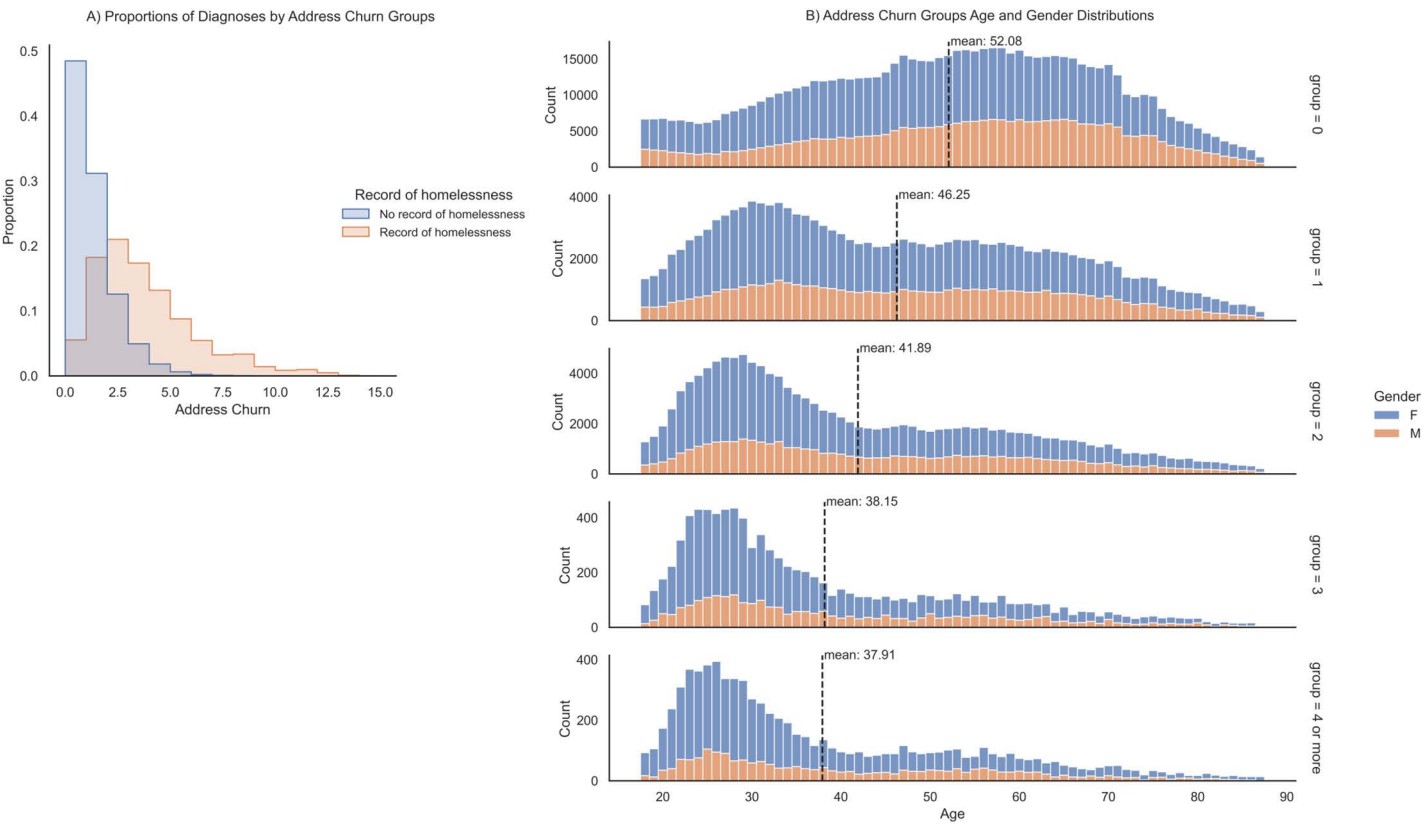

**Fig 1. Address churn distributions by homeless status and demographics.** A) Distributions of address churn by people without record of homelessness in the EHR and people with record of homelessness in the EHR. B) Distributions of age and sex per address churn group. Mean age of each group indicated by dotted black line.

significantly associated with churn group for young adults, adults, and seniors. Prevalence of psychiatric diagnoses increases as address churn increases starting with 27.7% of patients in the zero-address churn group compared to 65.2% in the four or more-address churn group. Pain and infectious diagnoses similarly increase across groups while cardiovascular diagnoses decrease (Fig 2).

Looking more granularly at the diagnoses per group, we find hypertension is the most common diagnosis for people with zero address churn with a prevalence of 30.6%, followed by vitamin D deficiency and hyperlipidemia. In the one-address churn group, vitamin D deficiency and hypertension are the top two most common diagnoses (27.7% and 27.4% respectively), with anxiety and fear-related diagnoses ranking sixth at 21.6%. Anxiety and fear-related disorders are the second most common diagnosis in the two-address churn group (28.2%), the most common diagnosis for the three-address churn group (38.1%) and reaching a peak in the four or more-address churn group (45.4%). Anhedonia, depressive disorders, and pain-related disorders are notably frequent in the latter group (S1 Table).

## Logistic regression model

We find that address churn is significantly associated with ever experiencing homelessness before and after adjustment with other demographic variables with increased odds of 1.44 95% CI [1.42 - 1.47] for each additional residential address change. Male gender (OR 3.43 95% CI [3.09 - 3.79]), EHR recorded self-identified black or African American race (OR 3.42 95% CI [3.03 - 3.84]), insurance status especially Medicaid (OR 22.05 95% CI [18.48 - 26.30]),

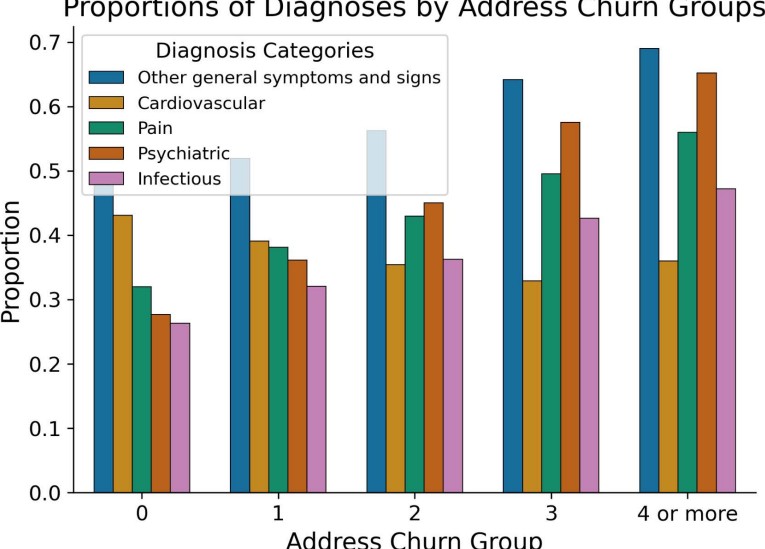

**Fig 2. Diagnostic profiles of address churn groups.** Proportions of diagnosis groups: other general symptoms and signs, cardiovascular, pain, psychiatric, and infectious per address churn groups.

previous diagnosis of severe mental illness (OR 5.77 95% CI [5.18 - 6.43]), previous diagnosis of substance abuse (OR 4.19 95% CI [3.77 - 4.65]), and deprivation index (OR 1.30 95% CI [1.23 - 1.37]) are also found to be significantly associated with record of experiencing homelessness (Table 2).

## Geospatial analysis of churn and ADI

The plot of mean patient address churn per zip code shows that the areas in central Chicago tend to have patients residing with higher mean address churn, with River North (zip code 60654) having the highest average patient address churn of 1.55 but low ADI 0.26 On the other hand, areas in southwest and west Chicago have the highest ADI. In particular, the neighborhoods of Englewood and Washington Park (zip code 60621) in the southwest region of Chicago have ADI of 0.69 and address churn of 1.17 and South Lawndale (zip code 60623) in the west has ADI of 0.68 and address churn of 0.87 (Fig 3A and 3B).

## Discussion

### Overview

In our study we find that address churn is significantly associated with having experienced homelessness with increased odds of 1.44 [95% CI 1.42 - 1.47]. Other variables associated with ever experiencing homelessness include self-reported race, with EHR recorded race as Black or African American compared to recorded race of White having the greatest odds ratio of 3.42 (95% CI [3.03 - 3.84]), insurance status with Medicaid compared to commercial insurance having greatest odds ratio of 22.05 (95% CI [18.48 - 26.3]), diagnosis of SMI, diagnosis of substance abuse, and area deprivation index. We observe demographic variability among the different address churn groups with regards to age, housing status, and neighborhood. We also show diagnostic differences across the levels of address churn with greater address churn tending towards greater frequency of anxiety, mood-related diagnoses, and chronic pain, while lower

**Table 2. Univariate and multivariate logistic regression odd ratio and 95% CI.**

| Variable | Univariate | | Multivariate | |
|---|---|---|---|---|
| | Odds Ratio [95% CI] | p-value | Odds Ratio [95% CI] | p-value |
| Age | 0.99 [0.99 - 1.0] * | <0.001 | 1.00 [0.99 - 1.0] | 0.335 |
| Sex (Reference Female) | | | | |
| Male | 3.34 [3.04 - 3.66] * | <0.001 | 3.43 [3.09 - 3.79] * | <0.001 |
| Race (Reference White) | | | | |
| Black or African American | 10.64 [9.71 - 11.66] * | <0.001 | 3.42 [3.03 - 3.84] * | <0.001 |
| Asian | 0.70 [0.5 - 0.98] * | 0.040 | 1.08 [0.76 - 1.53] | 0.679 |
| AI/AN/NHPI | 2.11 [1.0 - 4.44] | 0.050 | 1.68 [0.77 - 3.64] | 0.19 |
| Other | 0.36 [0.3 - 0.44] * | <0.001 | 0.51 [0.41 - 0.63] * | <0.001 |
| Churn | 1.66 [1.64 - 1.69] * | <0.001 | 1.44 [1.42 - 1.47] * | <0.001 |
| Insurance (Reference Commercial Insurance) | | | | |
| Medicaid | 73.16 [61.85 - 86.55] * | <0.001 | 22.05 [18.48 - 26.30] * | <0.001 |
| Medicare | 7.63 [6.4 - 9.11] * | <0.001 | 6.03 [4.95 - 7.36] * | <0.001 |
| Other | 4.00 [3.14 - 5.09] * | <0.001 | 5.66 [4.43 - 7.22] * | <0.001 |
| Area Deprivation Index | 1.92 [1.85 - 1.99] | 0.069 | 1.30 [1.23 - 1.37] * | <0.001 |
| Severe Mental Illness | 29.94 [27.36 - 32.77] * | <0.001 | 5.77 [5.18 - 6.43] * | <0.001 |
| Substance Abuse Diagnosis | 19.53 [17.87 - 21.36] * | <0.001 | 4.19 [3.77 - 4.65] * | <0.001 |

CI = Confidence interval.

AI/AN/NHPI = American Indian/Alaskan Native/Native Hawaiian and Pacific Islander.

*= P-value <0.05 considered significant.

churn groups tend to have higher frequency of "diseases of affluence" in the zero-address churn group.

## Association between address churn and homelessness

Our finding that address churn is statistically significantly associated with homelessness supports our hypothesis that it is a marker for housing instability, as housing status exists on a continuum with the most severe instability being homelessness [4,46]. To our knowledge our study is the first to explore address churn in the EHR as a measure of housing instability [22,27,28,36,37]. We propose that address churn, a variable that is already recorded in EHRs, can contribute to improved screening for individuals experiencing housing instability or individuals at risk of homelessness. Address churn shows a significant effect in the univariate and multivariate models, but the effect size is attenuated in the full model, suggesting that, although churn is an important predictor on its own, its contribution is moderated by other SDOH. Existing literature similarly supports the idea that a multitude of external factors including mental health, income, and neighborhood safety may play a role in homelessness [47–49]. As these processes occur, an individual may experience unstable housing requiring frequent address changes [50]. Recognizing housing instability is not just relevant for housing interventions [51,52]. It is also a public health issue as it has been shown to exacerbate chronic conditions like heart failure, hypertension, and COPD, heighten the risk of infectious diseases and violence, and incur higher healthcare costs due to longer hospital stays and more emergency visits [12,14,16,17,53–56]. Understanding address churn in the context of other demographic factors can improve screening in clinics and contribute to improving health outcomes associated with homelessness and housing instability.

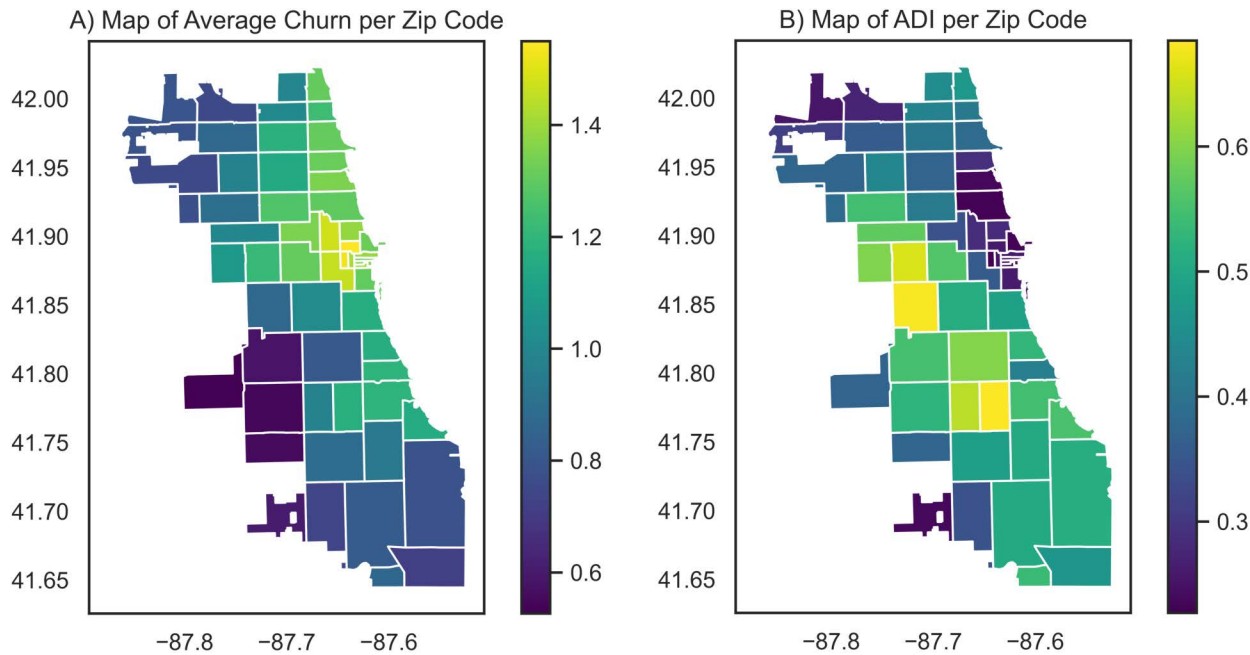

**Fig 3. Geospatial distributions of address churn and ADI in Chicago.** A) Average address churn per zip code in Chicago. B) ADI per zip code in Chicago.

## Describing and characterizing address churn

As we attempt to define a threshold for high churn, we find there is no consensus for what it means to have high residential mobility. The U.S. Census finds that on average, Americans move 11.7 times in their lifetimes, while other studies find that three or four residential moves regardless of timespan may be considered high mobility [57,58]. Our findings add to the literature by taking time interval into consideration. We observe patients who have no record of homelessness have average address churn of less than one in a three-year period, while patients who have history homelessness have an average churn of about two in a three-year interval.

To better characterize churn, we observe demographic differences with varying levels of churn. Most patients included in the study do not move over three years. This indicates a general stability in housing for much of the patient population. Patients with a record of homelessness have a statistically significant higher mean churn compared to those without, highlighting the instability faced by individuals with a history of homelessness. The higher address churn among these patients suggests ongoing housing insecurity, which further supports our main hypothesis that address churn is an early marker of homelessness. A one-way ANOVA reveals a statistically significant difference in age distribution between the address churn groups. Younger individuals are more likely to experience higher address churn compared to older individuals. This may be due to various factors such as employment instability, educational pursuits, or lack of established housing. Previous literature also supports the idea of "ageing-in-place" for older adults in metropolitan areas and that moves are generally motivated by major life events [59,60].

## Diagnosis and association with churn

Each group of address churn shows a distinct diagnostic profile. Address churn and diagnosis, stratified by age, are significantly associated. Meaning, while young adults have greater prevalence of mood disorders compared to older adults this does not fully explain the differences in

diagnoses [61]. We observe that people in higher address churn groups have greater frequencies of psychiatric diagnoses, specifically, anxiety and fear-related disorders. This may be due to the stressors associated with residential mobility which is often reactive to situations like evictions and living conditions [62–65]. This finding highlights the potential mental health risks associated with high address churn and underscores the need for targeted mental health support for these populations. The higher frequency of pain-related disorders in the higher churn groups may be attributed to pain being more poorly controlled, exacerbated by lack of consistent access to healthcare and poor living conditions [66,67]. This finding emphasizes the importance of ensuring continuous healthcare access and support for individuals with high address churn. Overall, the common diagnoses in the high churn groups are like diagnoses seen in people experiencing homelessness [15,53,68,69]. The lower churn groups have greater prevalence of diseases such as hypertension and hyperlipidemia, often affecting older and wealthier individuals. Alternatively, it could indicate underdiagnosis or lack of access to healthcare services among those with high address churn.

## Geospatial patterns of churn

Areas of high churn do not necessarily overlap with areas of high deprivation, which may be explained by the patterns of immobility seen in areas of lower socioeconomic status. Reports show that generally families in lower socioeconomic standing report living in the same residence until an external factor like extreme violence forces them to move elsewhere [62,70]. River North, a downtown Chicago neighborhood with high housing costs, has the highest address churn and corresponds to areas of higher homelessness per the Chicago Point-in-Time estimate and may be driving up the average churn of the area [71].

## Limitations

Our study has several limitations and may be subject to several potential biases. Measurement bias could arise from inaccuracies in recording address changes or diagnoses in the medical records. Measurement bias may also arise as there is no standardized method of identifying homelessness. Our method of using address and ICD-10 codes relies on the clinician's screening ability and the patient's honesty. As a result, ours is likely a conservative estimate of the number of individuals experiencing homelessness in the EHR. Information bias may occur as the NM-EDW inherently possesses limitations that may affect the validity of research findings including issues related to documentation errors and missing data. We attempt to mitigate these biases by standardizing addresses prior to analysis and by validating our data through chart review. The generalizability of our findings may be limited by the specific characteristics of our study population. Our study took place during the COVID-19 pandemic which may skew results due to the volatility of housing and healthcare usage during this time [72,73]. We attempt to temper the effects of COVID-19 by including three additional years (2021-2024) to improve generalizability of our findings. The population included in our study may not be fully representative of Chicago as we pulled patients from the Northwestern health system, which has an overlapping catchment area with other large health systems. The population is primarily patients from an urban healthcare system which may not be generalizable to rural settings. Future research should aim to replicate our study in diverse settings to enhance the generalizability of the results.

Several assumptions were made during our analysis. While this study simplifies cases of homelessness to binary categorization, we acknowledge that a broad range of housing instability and homelessness exist, and more granular exploration may be necessary but are beyond the scope of this paper. We recognize that housing is complex and residential address churn may be one dimension among many [4]. We acknowledge that residential mobility can reflect

positive change (i.e., moving towards more stable housing) but literature supports that it is more frequently a symptom of housing instability [74–76].

## Conclusion

In conclusion, our study finds a statistically significant association between address churn and homelessness, which highlights a potential avenue for administrative screening to detect signs of housing instability with variables already routinely captured in EHRs. Traditional indicators including ICD-10 codes and address records may not be as effective at identifying the nuances in housing instability or in identifying individuals at risk before they experience homelessness. We find higher prevalence of anxiety, mood disorders, and chronic pain among those with higher churn, indicating a link between psychosocial stressors and health. While most patients have a churn of zero, we find that people with record of homelessness and young adults tend to have higher churn. Knowing who and where high address churn occurs can help create targeted screening and policy changes. Standardized screening protocols can be supported through collaboration between healthcare providers, social services, and housing agencies. Policymakers should advocate for stable housing as a health determinant and allocate resources to housing stability initiatives. Our findings underscore a potential measure to facilitate the integration of housing stability screening into healthcare systems to identify at-risk individuals early and provide timely interventions.

## Supporting information

**S1 Table. Top ten diagnoses and prevalence per address churn group.**
(DOCX)

## Acknowledgments

This work was supported in part by the Northwestern Medicine Enterprise Data Warehouse.

## Author contributions

**Conceptualization:** Janet Song, Yu Deng, Lacey Gleason, Abel Kho.

**Data curation:** Janet Song, Yu Deng, Yuyang Yang.

**Formal analysis:** Janet Song, Yuyang Yang, Abel Kho.

**Investigation:** Janet Song, Yu Deng, Yuyang Yang, Lacey Gleason, Abel Kho.

**Methodology:** Yuyang Yang, Lacey Gleason, Abel Kho.

**Project administration:** Janet Song, Yu Deng, Lacey Gleason, Abel Kho.

**Visualization:** Janet Song.

**Writing – original draft:** Janet Song, Abel Kho.

**Writing – review & editing:** Janet Song, Yu Deng, Yuyang Yang, Lacey Gleason, Abel Kho.

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
