## [Decision Letter · Decision Letter 0]

25 Jun 2024

PONE-D-24-20931Change in Address in Electronic Health Records as an Early Marker of HomelessnessPLOS ONE

Dear Dr. Song,

Thank you for submitting your manuscript to PLOS ONE. After careful consideration, we feel that it has merit but does not fully meet PLOS ONE’s publication criteria as it currently stands. Therefore, we invite you to submit a revised version of the manuscript that addresses the points raised during the review process.

We look forward to receiving your revised manuscript.

Kind regards,

Asadullah Shaikh, Ph.D.

Academic Editor

PLOS ONE

Journal Requirements:

  "I have read the journal's policy and the authors of this manuscript have the following competing interests:

Dr. Abel Kho is an advisor for Datavant."

Reviewers' comments:

Reviewer's Responses to Questions

**Comments to the Author**

1. Is the manuscript technically sound, and do the data support the conclusions?

Reviewer #1: Yes

Reviewer #2: Yes

2. Has the statistical analysis been performed appropriately and rigorously? 

Reviewer #1: Yes

Reviewer #2: Yes

3. Have the authors made all data underlying the findings in their manuscript fully available?

Reviewer #1: Yes

Reviewer #2: No

4. Is the manuscript presented in an intelligible fashion and written in standard English?

Reviewer #1: Yes

Reviewer #2: Yes

5. Review Comments to the Author

Reviewer #1: REVIEWER’s REPORT

On the paper PONE-D-24-20931

Change in Address in Electronic Health Records as an Early Marker of Homelessness

by Janet Song, Yu Deng, Yuyang Yang, Lacey Gleason, Abel Kho,

In this paper the authors make a single-center cross-sectional study of EHR data (2018-2024) conducted at a US academic center with eleven sites across Chicago, including patients 18 years or older with at least three hospital encounters over three different years. The authors define address churn as the number of address changes recorded in the EHR corrected to three-year intervals and compare demographic and clinical characteristics of individuals with varying address churn, performing multivariable logistic regression to measure the association between people with a record of homelessness and address changes.

The results are new, correct and detailed. The paper is original and doesn’t contradict to ethical or policy issues, the question posed by authors is new and well defined, the methods used by authors are appropriate and well described, the data are sound and well controlled, the discussion and conclusions are well balanced, the title and abstract convey the obtained results, the writing is acceptable, the paper contains good scientific results.

The paper doesn’t require a revision.

Taking the above into consideration, I recommend the paper for publication, in PLOS ONE.

9.06.2024

Reviewer #2: ABSTRACT: The abstract is clear but briefly mentioning the statistical methods used will further enhance clarity. Also add a brief statement on the impact of the findings on public health or clinical practice

LITERATURE REVIEW: A section on Literature review is necessary. Though the Introduction is a bit clear about the context but it will be better to strengthen it with a more thorough review of existing literature on the use of EHR data for screening social determinants of health, and identifying gaps in previous studies. Conclude the section by highlighting the gap(s) that your study is addressing in clearer terms. Clearly state research questions or hypotheses at the end of the Introduction section. This will help frame the subsequent sections and guide the reader.

METHODS: Ensure the study design is explicitly stated as cross-sectional and clarify why this design is appropriate for your research questions. Provide more details about the Electronic Data Warehouse (EDW), such as its scope, the types of data it includes, and any limitations. Do a brief description of USPS Postal Addressing Standards and its process more explicitly and explain how it is good for your study. Detail the consultation with housing workgroup experts and any validation of the ICD-10 codes and address records used to identify homelessness. Consider adding specific statistical tests of stated hypotheses and how they help address the research questions. Also add a section on possible confounding variables and how you handled them.

RESULTS & DISCUSSIONS: The results section presents a lot of data but could be better structured with the view to helping the readers understand the context. You should clearly indicate which results are statistically significant and discuss the clinical relevance of these findings. Do a detailed interpretation of key findings of the research. Discuss how these findings compare to previous studies and what they add to the existing knowledge. Explain the potential biases and the generalizability of the findings, and any assumptions made during the analysis.

CONCLUSION: Summarize your key findings more explicitly and discuss their implications for healthcare practice and policy. Furthermore, discuss the actions that healthcare sectors, clinicians, and policymakers will need to take to strengthen the integration of housing stability screening in healthcare systems.

REFERENCES: Ensure all references are up-to-date and relevant. You will need to add more recent studies that are very relevant to your research.

LANGUAGE: Make sure that the manuscript is free of grammatical errors and follow a consistent style of writing. I advise you engage use of language editor. Also ensure each section flows logically into the next, with clear connections between the research question, methods, results, and discussion.

6. PLOS authors have the option to publish the peer review history of their article (what does this mean? ). If published, this will include your full peer review and any attached files.

**Do you want your identity to be public for this peer review?** For information about this choice, including consent withdrawal, please see our Privacy Policy .

Reviewer #1: **Yes: ** Alina Alb Lupas

Reviewer #2: **Yes: ** Boluwaji Ade Akinnuwesi

---

## [Author Response · Author response to Decision Letter 1]

2 Aug 2024

Our point-by-point response to reviewers and editors are typed out below and uploaded as a document labeled "Response to Reviewers".

Response to Reviewer #1

In this paper the authors make a single-center cross-sectional study of EHR data (2018-2024) conducted at a US academic center with eleven sites across Chicago, including patients 18 years or older with at least three hospital encounters over three different years. The authors define address churn as the number of address changes recorded in the EHR corrected to three-year intervals and compare demographic and clinical characteristics of individuals with varying address churn, performing multivariable logistic regression to measure the association between people with a record of homelessness and address changes.

The results are new, correct and detailed. The paper is original and doesn’t contradict to ethical or policy issues, the question posed by authors is new and well defined, the methods used by authors are appropriate and well described, the data are sound and well controlled, the discussion and conclusions are well balanced, the title and abstract convey the obtained results, the writing is acceptable, the paper contains good scientific results.

The paper doesn’t require a revision.

Taking the above into consideration, I recommend the paper for publication, in PLOS ONE.

Thank you for your thorough and positive review of our manuscript. We are delighted to hear that you found our study to be original, well-defined, and methodologically sound. We appreciate your recognition of the novelty and detail of our results, as well as your acknowledgment of the balanced discussion and conclusions. We are grateful for your recommendation for publication in PLOS ONE and are excited about the opportunity to contribute to the scientific community through this platform.

Response to Reviewer #2

Thank you for your detailed and constructive review of our manuscript. We appreciate your feedback and insightful suggestions. Your comments have provided valuable guidance for enhancing the clarity and comprehensiveness of our paper. We will address each of your points in detail to ensure that our manuscript meets the highest standards of quality and rigor.

ABSTRACT: The abstract is clear but briefly mentioning the statistical methods used will further enhance clarity. Also add a brief statement on the impact of the findings on public health or clinical practice.

We added a statement mentioning the statistical methods (lines 30-34) and a statement on the impact of our findings on public health to our abstract (lines 48-50).

Lines 30-34: “We compare demographic and clinical characteristics of individuals with varying address churn with the student T-test to look at distribution of address churn for patients with and without record of homelessness, ANOVA to evaluate the distribution of ages for different levels of churn, and the chi-square test to evaluate for association between churn and clinical diagnoses.”

Lines 48-50: “Our findings can help public health policy makers in targeting vulnerable populations at risk of homelessness with multiple health comorbidities for interventions such as housing first.”

LITERATURE REVIEW: A section on Literature review is necessary. Though the Introduction is a bit clear about the context but it will be better to strengthen it with a more thorough review of existing literature on the use of EHR data for screening social determinants of health, and identifying gaps in previous studies. Conclude the section by highlighting the gap(s) that your study is addressing in clearer terms. Clearly state research questions or hypotheses at the end of the Introduction section. This will help frame the subsequent sections and guide the reader.

We added 9 additional references regarding the usage of EHR data for screening of social determinants of health from 2023 and 2024 for a more thorough and updated literature review. Throughout the paragraph we highlight pitfalls of existing methods of EHR screening and conclude the paragraph by summarizing the gap our study is addressing (lines 69-93).

We have added our main hypothesis and research questions to the last paragraph of our Introduction (lines 94-104).

METHODS: Ensure the study design is explicitly stated as cross-sectional and clarify why this design is appropriate for your research questions. Provide more details about the Electronic Data Warehouse (EDW), such as its scope, the types of data it includes, and any limitations. Do a brief description of USPS Postal Addressing Standards and its process more explicitly and explain how it is good for your study. Detail the consultation with housing workgroup experts and any validation of the ICD-10 codes and address records used to identify homelessness. Consider adding specific statistical tests of stated hypotheses and how they help address the research questions. Also add a section on possible confounding variables and how you handled them.

We added a sentence explicitly stating the study is cross-sectional and why it is appropriate (lines 107-108).

Lines 107-108: “We utilize a retrospective cross-sectional study design to get a baseline snapshot of address churn…”

We have provided more details about the Electronic Data Warehouse (EDW), including its scope, and the types of data it includes (Lines 109-114). We have included limitations in the Limitations section under the Discussion (Lines 379-381).

Lines 109-114: “The NM-EDW is a data repository that includes clinical and research data from sites within Northwestern University School of Medicine and Northwestern Medicine Healthcare (NMHC) for greater than 10 million patients spanning from 1970 to the present. It contains demographic data (age, gender, race, address), clinical data (diagnoses, medications, encounters, procedures), and outcomes extracted from the Northwestern Medicine EHR.”

We added in the Methods section a description of USPS Postal Addressing standards, the process of capitalizing addresses and mapping spelling variations to standards, and how it helped cut down on redundant addresses for more accurate measurements (lines 133-142).

We added a description of the collaboration with housing workgroup experts to develop set of criteria to find patients with record of homelessness. We also explained our validation with chart review and how we narrowed our criteria (lines 144-161).

We have clarified how the statistical tests helped address each hypothesis and research question (lines 173-184, lines 214-215, and lines 223-228).

We have added a section on possible confounding variables above the Statistical Analysis section (lines 205-212).

RESULTS & DISCUSSIONS: The results section presents a lot of data but could be better structured with the view to helping the readers understand the context. You should clearly indicate which results are statistically significant and discuss the clinical relevance of these findings. Do a detailed interpretation of key findings of the research. Discuss how these findings compare to previous studies and what they add to the existing knowledge. Explain the potential biases and the generalizability of the findings, and any assumptions made during the analysis.

We have clarified the results section by explaining how each result is a response to our hypothesis and research questions to provide more context to the readers (lines 231, 247, 268). We have included asterisks in Table 2 for statistically significant results and have ensured that results that were statistically significant presented in the text were also emphasized.

We added a more detailed interpretation of key findings and how these compare to previous studies and that we are adding to existing knowledge a new variable for SDOH screening that has not previously been considered. (lines 305-372).

We reorganized the limitations section to make it more clearly structured in addressing potential biases, generalizability, and assumptions (lines 373-399).

CONCLUSION: Summarize your key findings more explicitly and discuss their implications for healthcare practice and policy. Furthermore, discuss the actions that healthcare sectors, clinicians, and policymakers will need to take to strengthen the integration of housing stability screening in healthcare systems.

We have added to the Conclusion section by more explicitly stating our key findings and their implications for healthcare practice and policy. We added how clinicians and policy makers can strengthen the integration of housing stability screening to the end of the conclusion section (lines 401-416).

REFERENCES: Ensure all references are up-to-date and relevant. You will need to add more recent studies that are very relevant to your research.

We reviewed our references ensuring they are up-to-date and relevant. We added 17 additional references from 2023-2024 that are relevant to our research.

LANGUAGE: Make sure that the manuscript is free of grammatical errors and follow a consistent style of writing. I advise you engage use of language editor. Also ensure each section flows logically into the next, with clear connections between the research question, methods, results, and discussion.

We reviewed our manuscript several times to ensure it is free of grammatical errors, follows a consistent style of writing, and clear connections between the different sections. We used Microsoft Editor, a language editor built into Microsoft Word.

Additional Requirements

1. Please ensure that your manuscript meets PLOS ONE's style requirements, including those for file naming. The PLOS ONE style templates can be found at :

We have updated the title page of our manuscript to fit with PLOS ONE’s style requirements: Article title written in sentence case, numeric author affiliations in order, titles removed from author byline, removal of abbreviation from affiliations, removal of physical address for corresponding author, and inclusion of corresponding author’s initials.

We have updated the main body of our manuscript formatting to fit with PLOS ONE’s style requirements with the following: use of level 1 heading for all major sections, use of level 2 heading for sub-sections, use of sentence case for all headings, brackets for in-text citations, and references updated per PLOS ONE format.

"I have read the journal's policy and the authors of this manuscript have the following competing interests:

Dr. Abel Kho is an advisor for Datavant."

We have updated our Competing Interests statement to clarify that current conflicts of interest do not alter adherence to PLOS ONE policies and included it in the cover letter worded as below:

“I have read the journal's policy and the authors of this manuscript have the following competing interests:

Dr. Abel Kho is an advisor for Datavant.

This does not alter our adherence to PLOS ONE policies on sharing data and materials."

a. If there are ethical or legal restrictions on sharing a de-identified data set, please explain them in detail (e.g., data contain potentially identifying or sensitive patient information, data are owned by a third-party organization, etc.) and who has imposed them (e.g., a Research Ethics Committee or Institutional Review Board, etc.). Please also provide contact information for a data access committee, ethics committee, or other institutional body to which data requests may be sent.

b. If there are no restrictions, please upload the minimal anonymized data set necessary to replicate your study findings to a stable, public repository and provide us with the relevant URLs, DOIs, or accession numbers. For a list of recommended repositories, please see

We have clarified our data sharing statement, that data contains potentially identifying patient information, and that data are accessible through the Northwestern Medicine Enterprise Data Warehouse with contact information worded as below. We also include the Northwestern Medicine Enterprise Data Warehouse in acknowledgements.

“Data cannot be shared publicly due to legal restrictions as data contain potentially identifying patient information. Data are available from the Northwestern University Northwestern Medicine Enterprise Data Warehouse (NMEDW) (contact via fsm-research@northwestern.edu) for researchers who meet the criteria for access to confidential data.”

We have updated the “Methods” section of our manuscript to include our full ethics statement to include the full name of the IRB who approved our study and that we obtained waiver of informed written or verbal consent (lines 124-128).

“The study protocol was approved by the Northwestern University Institutional Review Board. Waiver of informed consent was obtained for this study because the research involved no more than minimal risk to patients as this study uses pre-existing data, the waiver will not adversely affect the rights and welfare of participants, and it would not be practicable to carry out the research without the waiver.”

---

## [Decision Letter · Decision Letter 1]

27 Sep 2024

PONE-D-24-20931R1Change in Address in Electronic Health Records as an Early Marker of HomelessnessPLOS ONE

Dear Dr. Song,

Thank you for submitting your manuscript to PLOS ONE. After careful consideration, we feel that it has merit but does not fully meet PLOS ONE’s publication criteria as it currently stands. Therefore, we invite you to submit a revised version of the manuscript that addresses the points raised during the review process.

We look forward to receiving your revised manuscript.

Kind regards,

Asadullah Shaikh, Ph.D.

Academic Editor

PLOS ONE

Reviewers' comments:

Reviewer's Responses to Questions

**Comments to the Author**

1. If the authors have adequately addressed your comments raised in a previous round of review and you feel that this manuscript is now acceptable for publication, you may indicate that here to bypass the “Comments to the Author” section, enter your conflict of interest statement in the “Confidential to Editor” section, and submit your "Accept" recommendation.

Reviewer #3: All comments have been addressed

Reviewer #4: (No Response)

Reviewer #5: All comments have been addressed

2. Is the manuscript technically sound, and do the data support the conclusions?

Reviewer #3: Yes

Reviewer #4: Partly

Reviewer #5: Yes

3. Has the statistical analysis been performed appropriately and rigorously? 

Reviewer #3: Yes

Reviewer #4: No

Reviewer #5: Yes

4. Have the authors made all data underlying the findings in their manuscript fully available?

Reviewer #3: No

Reviewer #4: No

Reviewer #5: (No Response)

5. Is the manuscript presented in an intelligible fashion and written in standard English?

Reviewer #3: Yes

Reviewer #4: No

Reviewer #5: Yes

6. Review Comments to the Author

Reviewer #3: (No Response)

Reviewer #4: Inadequate Introduction: Introduction should include proper background of the research area and research topic as well as the works from state of the arts.

Graphics and Visual Aids: All of the figures and tables in the manuscript could be enhanced for clarity and better understanding.

Formatting and Structure: There are many inconsistencies in the formatting and structure of the manuscript that could be streamlined for a more professional presentation. Authors uses the same heading two times. “Diagnosis data” line 224 and “Diagnoses” in line 307

Related works section missing

In overall, this manuscript looks like a technical web report

Reviewer #5: (No Response)

7. PLOS authors have the option to publish the peer review history of their article (what does this mean? ). If published, this will include your full peer review and any attached files.

**Do you want your identity to be public for this peer review?** For information about this choice, including consent withdrawal, please see our Privacy Policy .

Reviewer #3: No

Reviewer #4: **Yes: ** Awais Ahmad

Reviewer #5: **Yes: ** Muhammad Ahmad Pasha

---

## [Author Response · Author response to Decision Letter 2]

6 Nov 2024

1. Inadequate Introduction

Reviewer Comment: Introduction should include proper background of the research area and research topic as well as the works from state of the arts.

Response: Thank you for your feedback. We have revised the introduction to provide a more comprehensive background of the research area and topic. Specifically, we provided a broader historical context for our research topic and incorporated state of the art of general homelessness research in lines 55-58 and lines 62-72, respectively. Please see the related works section to get a more comprehensive literature review of housing screening in electronic health records.

2. Graphics and Visual Aids

Reviewer Comment: All of the figures and tables in the manuscript could be enhanced for clarity and better understanding.

Response: We appreciate your suggestion. We have reviewed our figures and processed them through PACE to best adhere to PLOSOne publication standards. We also encourage you to click the links in the PDFs as the figures in the PDF have lower resolution than the actual image accessed through the link. Tables have been reviewed to ensure numbers are clear and accurate.

3. Formatting and Structure

Reviewer Comment: There are many inconsistencies in the formatting and structure of the manuscript that could be streamlined for a more professional presentation. Authors uses the same heading two times. “Diagnosis data” line 224 and “Diagnoses” in line 307.

Response: Thank you for bringing this point of confusion to our attention. We have thoroughly reviewed and corrected the formatting and structure of the manuscript. The headings have been adjusted so the reader may better understand the content contained within each header. In line 203 we have changed “Diagnosis data” to be “Processing diagnosis data” to clarify that we are discussing our methodology of how we processed the diagnosis data. In line 262, we have changed the header from “Diagnoses” to “Diagnostic characteristics of address churn” to clarify that these are the results of our analyses looking at proportions of diagnoses per address churn group as displayed in Figure 2 and looking at top ten most frequent diagnoses per group. This header now parallels the preceding header in line 246, “Demographic characteristics of churn”.

We have additionally included headers in the discussion section to help guide readers (lines 310, 323, 344, 366, 385, and 394). We also restructured the discussion section to better convey our main findings how they support our hypothesis. With the exception of the “Association between address churn and homelessness” section, we have also reordered the discussion section to better parallel the order of the results section in lines 324-344.

We hope these changes help with making the overall presentation streamlined for a more professional appearance.

4. Related Works Section Missing

Reviewer Comment: Related works section missing.

Response: Thank you for your feedback, while there is not a separate related works section, we have a detailed review of related works already included in the introduction. We now include a heading in our paper highlighting the “Related works” section to improve the clarity. The introduction has been edited as discussed in item 1 “Inadequate Introduction” so that it includes a broader overview on the research matter and the relevant related works on housing screening is in a separate “Related works” section on lines 80-107.

5. Overall Presentation

Reviewer Comment: In overall, this manuscript looks like a technical web report.

Response: We appreciate your candid feedback. We have made significant revisions as detailed above to ensure that the manuscript adheres to academic standards and does not resemble a technical web report. These changes include improving the narrative flow, enhancing the academic tone, and ensuring consistency in formatting and style.

---

## [Decision Letter · Decision Letter 2]

20 Jan 2025

Change in Address in Electronic Health Records as an Early Marker of Homelessness

PONE-D-24-20931R2

Dear Dr. Song,

We’re pleased to inform you that your manuscript has been judged scientifically suitable for publication and will be formally accepted for publication once it meets all outstanding technical requirements.

Kind regards,

Asadullah Shaikh, Ph.D.

Academic Editor

PLOS ONE

Additional Editor Comments (optional):

Reviewers' comments:

Reviewer's Responses to Questions

**Comments to the Author**

1. If the authors have adequately addressed your comments raised in a previous round of review and you feel that this manuscript is now acceptable for publication, you may indicate that here to bypass the “Comments to the Author” section, enter your conflict of interest statement in the “Confidential to Editor” section, and submit your "Accept" recommendation.

Reviewer #4: All comments have been addressed

Reviewer #6: All comments have been addressed

2. Is the manuscript technically sound, and do the data support the conclusions?

Reviewer #4: Yes

Reviewer #6: Yes

3. Has the statistical analysis been performed appropriately and rigorously? 

Reviewer #4: Yes

Reviewer #6: Yes

4. Have the authors made all data underlying the findings in their manuscript fully available?

Reviewer #4: No

Reviewer #6: Yes

5. Is the manuscript presented in an intelligible fashion and written in standard English?

Reviewer #4: No

Reviewer #6: Yes

6. Review Comments to the Author

Reviewer #4: I am satisfied with the answers of the raised comments.......................................................................................................................................................................................................................................................................................................................................

Reviewer #6: (No Response)

7. PLOS authors have the option to publish the peer review history of their article (what does this mean? ). If published, this will include your full peer review and any attached files.

**Do you want your identity to be public for this peer review?** For information about this choice, including consent withdrawal, please see our Privacy Policy .

Reviewer #4: **Yes: ** Awais Ahmad

Reviewer #6: No

---

## [Editor Report · Acceptance letter]

PONE-D-24-20931R2

PLOS ONE

Dear Dr. Song,

I'm pleased to inform you that your manuscript has been deemed suitable for publication in PLOS ONE. Congratulations! Your manuscript is now being handed over to our production team.

Kind regards,

on behalf of

Prof. Asadullah Shaikh

Academic Editor

PLOS ONE
